# Manual Correction of Voxel Misclassifications in Mesiotemporal Structures Does Not Alter Brain–Behavioral Results in an Episodic Memory Task

**DOI:** 10.3390/jcm10214869

**Published:** 2021-10-22

**Authors:** Francina Hartmann, Julia Reinhardt, Christoph Stippich, Sabine Krumm

**Affiliations:** 1Memory Clinic FELIX PLATTER, University Department of Geriatric Medicine FELIX PLATTER, 4055 Basel, Switzerland; 2Division of Diagnostic and Interventional Neuroradiology, University Hospital Basel, University of Basel, 4031 Basel, Switzerland; Julia.Reinhardt@usb.ch; 3Department of Neuroradiology, University Hospital Zurich, University of Zurich, 8091 Zurich, Switzerland; stippich.radiology@mailbox.org; 4Faculty of Medicine, University of Basel, 4056 Basel, Switzerland

**Keywords:** voxel-based morphometry, voxel misclassification, magnetic resonance imaging, Alzheimer’s disease, DARTEL, manual correction

## Abstract

Voxel-based morphometry (VBM) is an established method for assessing grey matter volumes across the brain. The quality of magnetic resonance imaging (MRI) and the chosen data preprocessing steps can affect the outcome of VBM analyses. We recognized a lack of publicly available and commonly used protocols, which indicates that standardized and optimized preprocessing protocols are needed. This paper focuses on the time- and resource-consuming manual correction of misclassifications of grey matter voxels in cortical structures important in Alzheimer’s dementia. A total of 126 individuals, including 63 patients with very early Alzheimer’s disease and 63 cognitively normal participants, received thorough neuropsychological testing and 3-Tesla MRI. Automated preprocessing of T1 MPRAGE images was performed, and misclassifications of grey matter voxels were manually identified and corrected. In a second run, the manual correction step was skipped. Multiple regression analyses using DARTEL in SPM8 were then conducted with the manually corrected and uncorrected sample, respectively. Manual correction of voxel misclassifications did not have a major impact on the correlation between episodic memory performance and structural brain imaging results. We conclude that, although performing all preprocessing steps remains the gold standard, skipping manual correction of voxel misclassifications is permitted when investigating populations on the Alzheimer’s disease spectrum.

## 1. Introduction

Structural magnetic resonance imaging (MRI) has become a standard tool in the diagnostic process of Alzheimer’s disease (AD) [1]. As it has been shown that cortical atrophy appears already in the early stages of AD [2,3], voxel-based morphometry (VBM) is an established method for detecting the particular neuropathological patterns of AD [4,5]. However, the quality of MRI preprocessing greatly varies between scientists, clinicians, and research sites. The direct impact of these different handlings on statistical outcome is not yet clear. In this study, we investigated the impact of manual correction of grey and white matter voxel misclassifications on brain–behavioral results.

Dysfunction of episodic memory belongs to the main characteristics of early AD. Performance on verbal episodic memory tests, such as encoding and recall of word lists [6,7,8], is associated with reduced volume of the hippocampal and entorhinal cortex in patients with mild cognitive impairment (MCI) and Alzheimer’s dementia [9,10]. Additionally, Hirni et al. [11] showed that medial perirhinal and entorhinal cortex functioning is conspicuous even 12 years before diagnosis of Alzheimer’s dementia. Furthermore, reduced episodic memory performance is correlated with reduced volume of the entorhinal cortex in cognitively normal older adults [12]. A well-established task for testing verbal episodic memory performance is the California verbal learning task (CVLT) [13,14], discriminating successfully between groups of normal aging, patients with amnestic MCI (aMCI), and patients with early Alzheimer’s dementia [15]. In addition to neuropsychological assessment, the gold standard for diagnosing neurocognitive disorders comprises medical screening, gait analysis, cerebrospinal fluid measurement, positron emission tomography, and MRI. The latter has become a widely used diagnostic tool in recent decades due to its diagnostic accuracy, cost-effectiveness, and simplicity of application. Amongst other quantitative MRI measures, VBM is suggested to be a useful technique for the early detection of AD [16,17,18]. Chételat et al. [19], for example, assessed in a longitudinal voxel-based approach structural changes in the brain of patients that converted from aMCI to Alzheimer’s dementia and compared them with non-converters. The annual grey matter loss in converters compared with non-converters was highest in the temporal pole, medial temporal regions, and in the cingulate gyrus as well as the prefrontal cortex. VBM is therefore able to detect earliest AD-related changes in brain structure. It uses voxel-wise comparison of local grey matter concentration and, thus, produces results with high regional specificity. As an advantage, this technique allows the investigation of anatomically unbiased brain volume changes across the whole brain without the prior definition of regions of interest [20,21].

High quality MRI and accurate data preprocessing are crucial for producing reliable VBM results. Every step of VBM analysis, from acquisition to statistical analysis including correction for multiple comparisons, involves methodological decisions that can have a relevant impact on the results [22]. Even preprocessing steps such as segmentation procedure [23], normalization protocol [24], and smoothing kernel [25,26,27] may influence outcomes [28]. For instance, there are various methods for the segmentation of voxels into different brain tissues (i.e., grey matter, white matter, and cerebrospinal fluid). Choosing the right probabilistic atlas is highly important for preventing inaccurate results due to brain anatomy that significantly differs from the template [29]. Among different VBM techniques the Statistical Parametric Mapping software (SPM8, Wellcome Institute of Cognitive Neurology, www.fil.ion.ucl.ac.uk (accessed on 19 October 2021)) plus High Diffeomorphic Anatomical Registration Through Exponentiated Lie Algebra (DARTEL) procedure [30] shows advantages over other used algorithms with, e.g., SPM2/SPM5/SPM8 or the FMRIB software library (FSL, http://www.fmrib.ox.ac.uk/fsl/ (accessed on 19 October 2021)) enabling better segmentation and normalization of images, especially in populations with deviant anatomy [31,32]. By creating a template based on the population under study, SPM8 plus DARTEL is supposed to allow more local, non-linear deformations than conventional VBM, and thus, the discriminative power of MRI measures for AD diagnosis can be enhanced [31,33].

Nevertheless, grey and white matter voxels can be misclassified during automated MRI preprocessing steps. Typically, matching of individual anatomy to a template requires deformations particularly in regions that are enlarged in elderly subjects. Thus, segmentation and classification failures as well as method-related differences are predominantly found in cortical regions in close vicinity to ventricles, major sulci, and fissures [31,34,35,36,37]. These regions need review and often manual correction of voxel misclassifications. However, freely accessible standard protocols for the handling of voxel misclassifications and preprocessing steps are missing. We therefore contacted members in the fields of neuroimaging, radiology, neurology, neurobiology, and psychiatry of aging (e.g., postdoctoral fellows, research assistants, assistants, and full professors) from eight universities and four clinics. We knew from personal contact or thought on the basis of their publications or information on their webpage that they apply VBM techniques. We aimed to include different countries and contacted institutions located in Switzerland, England, the United States of America, Canada, France, Germany, Czech Republic, and Italy to ask if they apply standard written preprocessing protocols to MRI scans before images are released for clinical or scientific interpretation. Figure 1 gives a summary of the received responses to our search for standardized preprocessing protocols in VBM analysis.

To our knowledge and in line with the responses we received, there is no available literature addressing the handling of misclassifications and no existing standard protocol for preprocessing steps before VBM analysis. Manual preprocessing steps are very time-consuming. As measured by a researcher with only few experiences in preprocessing MRI data (FH), the manual correction of voxel misclassifications (i.e., reviewing each slice separately) can take up to 3 h per participant. In clinical settings or for large research samples, this effort is not bearable. Thus, our aim for this study was to optimize the VBM preprocessing protocol without losing meaningful information. We were particularly interested to find out whether manual correction of grey/white matter voxel misclassifications substantially increased the quality of studies that addressed only cortical regions. We therefore investigated the direct impact of manual correction of grey/white matter voxel misclassifications on brain–behavioral results in cortical structures important for AD diagnosis, such as e.g., structures of the parahippocampal gyrus or hippocampal formation.

## 2. Materials and Methods

### 2.1. Participants

We included baseline data from a longitudinal study of 126 native Swiss-German or German speaking adults with complete neuropsychological and MRI data sets. All participants were recruited from the Memory Clinic FELIX PLATTER, Basel, Switzerland. Written informed consent was obtained from all individuals prior to participation. The study was approved by the local ethics committee (EKNZ: Ethikkommission Nordwest-und Zentralschweiz).

The sample consisted of 63 individuals with very early AD (AD group), of which 35 (16 male, 19 female; mean age: 73.80 ± 9.17 (SD) years; range: 51–90 years; mean MMSE score: 28.37) were diagnosed with aMCI due to AD according to DSM-IV [38] and Winblad et al. [39] criteria. Twenty-eight individuals were diagnosed with early Alzheimer’s dementia (11 male, 17 female; mean age: 78.00 ± 5.30 (SD) years; range: 64–87 years; mean MMSE score: 26.43) according to NINCDS-ADRDA [40] and DSM-IV criteria [38]. Patients not only received neuropsychological testing including informant questionnaires but also received medical screening, gait analyses, and magnetic resonance imaging. Some patients had additional positron emission tomography scans and/or investigation of cerebrospinal fluid measures. They were diagnosed in an interdisciplinary consensus conference at the Memory Clinic FELIX PLATTER. Additionally, 63 cognitively normal participants (NC group; 38 male, 25 female; mean age: 74.54 ± 6.53 (SD) years; range: 60–90 years; mean MMSE score: 29.25) were selected and matched to the AD group with regard to age and education (both *p* > 0.30, Table 1). They had undergone medical screening and extensive neuropsychological testing to ensure that they were cognitively healthy (i.e., neurologically and psychiatrically). Socio-demographic characteristics and MMSE scores for NC and AD participants are depicted in Table 1. As expected, the two groups differed significantly in their MMSE score only (*t*(86) = 6.20, *p* = 1.89 × 10^−8^).

### 2.2. MRI Measures

#### 2.2.1. MRI Acquisition

MRI scanning was performed on a 3-Tesla scanner (MAGNETOM Verio, Siemens, Erlangen, Germany) at the University Hospital Basel, Switzerland (T1-weighted 3D magnetization-prepared rapid acquisition gradient echo (MPRAGE); 12 channel head coil; inversion time = 1000 ms; repetition time = 2000 ms; echo time = 3.37 ms; flip angle = 8°; field of view = 256 × 256; acquisition matrix = 256 × 256 mm; voxel size = 1 mm isotropic).

#### 2.2.2. Preprocessing of Structural MRI

Preprocessing of the T1 MPRAGE images was performed using the DARTEL method [30] in the SPM8 software (Wellcome Institute of Cognitive Neurology, www.fil.ion.ucl.ac.uk (accessed on 19 October 2021)) implemented in MATLAB 2010 (Mathworks Inc., Sherborn, MA, USA). Images were reoriented manually by setting the anterior commissure at the origin of the three-dimensional Montreal Neurological Institute (MNI) coordinate system. After automatic segmentation of MRI images into grey matter, white matter, and cerebrospinal fluid volumes, misclassifications of grey matter were manually identified and corrected on each slice. The MPRAGE images were then segmented again, masking the misclassifications as described elsewhere [41], and then co-registered to the DARTEL template and normalized to MNI space, modulated and smoothed with 8 mm FWHM Gaussian kernel.

For the uncorrected MRI data set, steps were analogous with the exception that manual identification and correction of misclassifications was skipped.

Total intracranial volume (TIV; see Table 1) was calculated as the sum of grey matter, white matter, and cerebrospinal fluid volumes per participant using the get_totals65.m program (http://www.cs.ucl.ac.uk/staff/G.Ridgway/vbm (accessed on 19 October 2021)) running on SPM12 (Wellcome Institute of Cognitive Neurology, www.fil.ion.ucl.ac.uk (accessed on 19 October 2021)) implemented in MATLAB R2010b (Mathworks Inc., Sherborn, MA, USA).

### 2.3. California Verbal Learning Task

In the course of neuropsychological testing, all participants completed the German version of the CVLT [13,14] to assess verbal episodic memory. A list of 16 nouns (List A) was read aloud to the participants during 5 trials. Each trial was followed by an immediate free recall. After completion of all trials, an interference list (List B) was presented and recalled, followed by a short-delay free recall and cued recall of List A. Finally, a long-delay free recall and cued recall of the words from List A as well as a recognition task was conducted.

Two neuropsychological measures were used for the analyses: the sum of recalled words from trial 1–5 of List A (CVLT_1–5; immediate recall), and the number of recalled words in the delayed free recall of List A (CVLT_LDFR; long-delay recall). Age-, sex-, and education-corrected z-scores were calculated according to a normative sample [42,43]. Because we only had normed data for age up to 88 years, three participants (one NC and two aMCI participants) had to be classified one year younger than their actual age to calculate the z-score.

### 2.4. Statistical Analyses

Socio-demographic data analysis was done with R version 3.3.3 [44] (https://www.R-project.org/ (accessed on 19 October 2021)). Voxel-based whole-brain correlations were conducted separately for the corrected and uncorrected data set, using CVLT immediate and delayed free recall as variables of interest (covariate: TIV). We performed group independent (i.e., over all participants) multiple regressions in SPM12 (Wellcome Institute of Cognitive Neurology, www.fil.ion.ucl.ac.uk (accessed on 19 October 2021)) implemented in MATLAB R2010b (Mathworks Inc., Sherborn, MA, USA). Significant voxels were identified using a threshold of *p* < 0.001 for the correlation with CVLT performance and a cluster size of >10.

Neuroanatomical regions of peak voxels (family-wise error (FWE)-corrected *p* < 0.05) were assigned using the aal template in MRIcron [45] (http://people.cas.sc.edu/rorden/mricron/ (accessed on 19 October 2021)).

## 3. Results

### 3.1. Behavioral Data

The NC group recalled significantly more words in the immediate recall (*t*(124) = 13.06, *p* < 2.2 × 10^−16^) as well as the delayed free recall condition (*t*(124) = 15.13, *p* < 2.2 × 10^−16^) compared with the AD group.

### 3.2. Voxel-Based Morphometry

Brain–behavioral analyses showed significant correlations between immediate recall CVLT_1–5 scores and grey matter volume for the manually corrected as well as for the uncorrected MRI data. Table 2a illustrates that one single cluster in the middle occipital gyrus (−37, −87, 38) did not show up in both analyses. Peak voxels with no distinction between corrected and uncorrected data were located in hippocampus, precuneus, temporal gyrus, and occipital gyrus (all *p* < 0.007; Table 2a, Figure 2).

For the CVLT_LDFR scores, we found significant correlations with voxels in the hippocampus, amygdala, precuneus, parietal gyrus, frontal gyrus, and occipital gyrus. As illustrated in Table 2b, results differed in two clusters in the inferior parietal gyrus (−24, −66, 44) and the middle occipital gyrus (−30, −89, 22) (all *p* < 0.046; Figure 3).

Cluster differences of corrected and uncorrected MRI data are shown in Figure 4.

## 4. Discussion

Comparing studies that use quantitative MRI measures in AD populations is difficult, as no common preprocessing protocol is available. The methodological decisions are highly variable [46,47], and the reporting in scientific papers is not always sufficiently transparent [48]. We asked a total of twelve research facilities and clinics from eight different countries whether or not they apply standardized MRI preprocessing protocols and/or perform manual correction before running VBM analyses. Feedback was very scarce, such that reliable interpretation is difficult. However, responses suggest that most facilities do not use standardized MRI preprocessing protocols and even less perform manual correction. There is a need for standardized and optimized (i.e., easy to use and highly efficient) preprocessing protocols for MRI scans to enhance the interpretation and comparison of VBM results across different studies and improve clinical practicability. A first step in this direction is to evaluate the effect on brain–behavioral results if manual voxel corrections are omitted.

As expected and reported by many other authors before, we found that NCs performed significantly better than the AD group in the immediate as well as the delayed free recall of the CVLT. This indicates that the used sample fulfills the requirements for our study (i.e., the sample enables us to investigate cortical structures).

In the immediate as well as the delayed free recall condition, VBM analyses revealed significant associations with five clusters that showed identical peak voxels for manually corrected and uncorrected data. In each condition, peak voxel location of a sixth cluster differed only in the x-axis by 1 mm between the corrected and uncorrected data. Furthermore, one cluster showed up only in the corrected data of the immediate free recall condition (middle occipital gyrus, −37, −87, 38). In the delayed free recall condition, one cluster was significant in the corrected (inferior parietal gyrus, −24, −66, 44) and another one in the uncorrected data only (middle occipital gyrus, −30, −89, 22). We think that these differences between the manually corrected and uncorrected data are only marginal. Significant clusters were located in regions expected to be involved in episodic memory, such as the hippocampus, amygdala, and precuneus. Figure 4 shows that results between corrected and uncorrected MRI data differed mainly at the borders of identified clusters and there was no specific pattern regarding the varying cluster sizes of manually corrected versus uncorrected data (see Table 2). For some regions, significant clusters comprise more voxels in the corrected data, while others contain more voxels in the uncorrected data (clusters differ in a range from 105 until 2972 voxels). One reason for this observation is that the accuracy of automated voxel allocation to grey and white matter differs depending on brain region and diagnosis. In normal aging, for example, studies reported preservation of grey matter volume in structures such as the amygdala and hippocampus [49,50,51]. These are precisely the regions showing more voxels in the uncorrected data. Ashburn and Friston [20] noted that voxels sometimes contain a mixture of tissues and usually appear as grey matter when located at the interface between white matter and ventricles. As we combined images from the NC and AD groups, signal intensity could have affected the primary assignment of voxels to grey or white matter and thus might have manipulated the manual misclassification correction.

Multiple regression analysis was performed by combining the two groups NC and AD to illustrate a continuous range of behavioral functioning and neuroanatomic structural patterns. As we investigated MRI images from very early AD patients, our findings can only be generalized very cautiously to AD patients in later stages of the disease. However, our results suggest that manual correction of grey/white matter voxel misclassifications can be skipped without losing meaningful information, specifically when investigating cortical regions in very early AD subjects. Considering that manual correction is subjective and that even experienced raters usually do not make identical modifications, it might be advisable to avoid the error-prone manual correction and save time by using automated approaches, which would enhance comparability between studies.

Methodological aspects tend to influence VBM results and their reproducibility [23,24,28,33,47]. Our results are restricted to preprocessing protocols using DARTEL in SPM8 (Wellcome Institute of Cognitive Neurology, www.fil.ion.ucl.ac.uk (accessed on 19 October 2021)). Callaert et al. [31] discussed the advantage of normalization and segmentation using SPM8 plus DARTEL over the SPM5/SPM8 algorithm. The updated segmentation procedure implemented in the SPM8 release determines additional tissue classes to reduce misclassification and takes into account the problem of suboptimal probability maps and age effects on grey matter estimates. In combination with DARTEL, errors can be reduced by using a template optimized for the population under study [30,52]. In the case of AD, SPM8 plus DARTEL is highly recommended as it has been shown to display better results in populations with deviant anatomy than conventional VBM by SPM5/SPM8 [31,53]. Therefore, our finding that we did not detect a specific pattern in which manually corrected and uncorrected data differ, may be explained to some extent by the better performance of the DARTEL approach over older versions.

Standardized guidelines for structural MRI preprocessing, especially for the handling of voxel misclassifications, are not only missing for AD populations but also for other diseases. Possibly, our approach can serve as template for the development of preprocessing protocols in other conditions. Ideally, the same standard structural MRI preprocessing protocol can be applied to diverse conditions, e.g., for patients with AD, chronic pain, spinal degeneration, or drug addiction. However, one hurdle to establishing a heterogeneously valid approach may be that involved brain regions vary between diseases. These regions can be differently vulnerable to the occurrence of voxel misclassifications, which can influence the results. It is important to note that our results can only be transferred with great caution to other populations, brain regions, resolutions, or studies with different methodological approaches. We used the CVLT, an episodic memory task, to measure behavior. A generalization to behavior per se is not possible. Furthermore, validation of the results by replication in a larger sample is recommended.

## 5. Conclusions

Manual correction of grey/white matter voxel misclassifications in structural MRI did not have considerable impact on brain–behavior correlations for episodic memory in AD patients using DARTEL in SPM8. Skipping manual correction of voxel misclassifications is therefore legitimate when specifically investigating cortical regions in AD populations by using a 3-Tesla MRI scanner, although performing all preprocessing steps remains the gold standard. Importantly, the presented results are restricted to AD patients in an early stage. A follow up study investigating how manual correction of voxel misclassifications influences results in later stages (i.e., when atrophy is more prominent) should be performed. We assume that our study is meaningful, as we noticed that many institutions use no standardized MRI preprocessing protocols and/or do not perform any manual correction of voxel misclassifications. The lack of transparency regarding used preprocessing protocols hampers the comparison and reproducibility of study results. Therefore, public guidelines for efficient data preprocessing are needed.

## Figures and Tables

**Figure 1 jcm-10-04869-f001:**
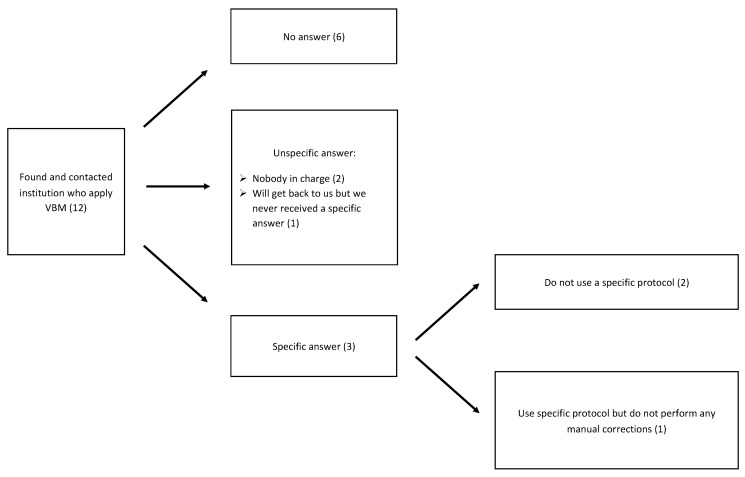
Flow diagram representing type of feedback received from universities and clinics regarding standardized preprocessing protocols in voxel-based morphometry analysis (number of answers in brackets).

**Figure 2 jcm-10-04869-f002:**
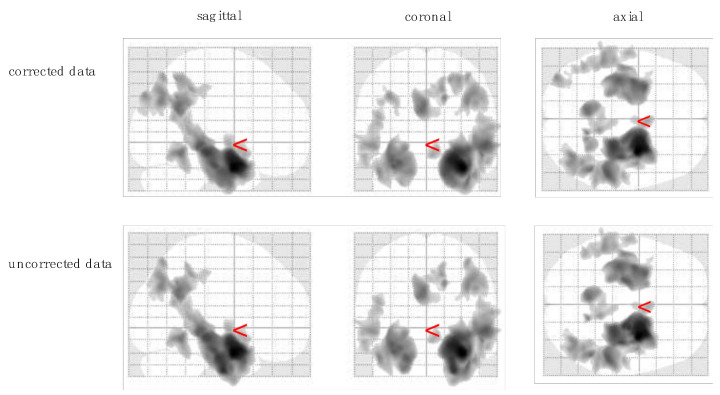
Results from voxel-based morphometry analysis showing grey matter clusters significantly correlating with CVLT immediate recall score (CVLT_1–5).

**Figure 3 jcm-10-04869-f003:**
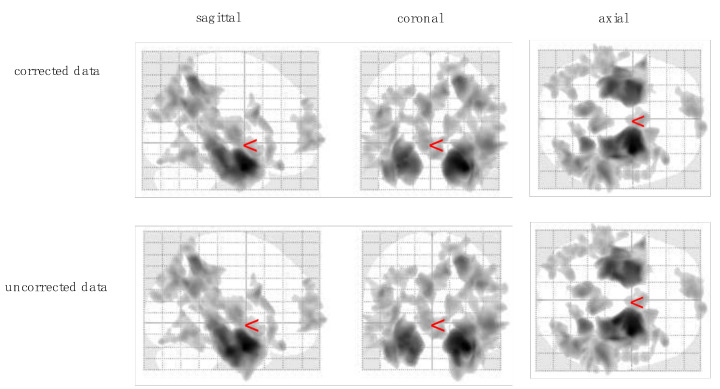
Results from voxel-based morphometry analysis showing grey matter clusters significantly correlating with CVLT long-delay free recall score (CVLT_LDFR).

**Figure 4 jcm-10-04869-f004:**
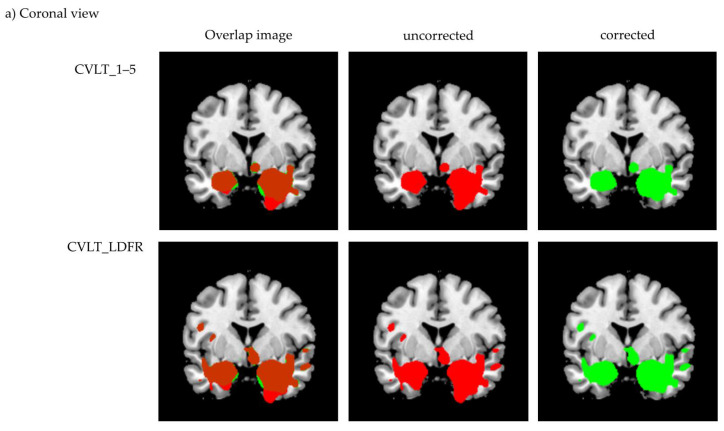
Comparison of voxel-based morphometry results for (**a**) coronal view and (**b**) sagittal view generated in MRIcron. Displayed is the overlap image, as well as separate images for the corrected (green) and uncorrected (red) data.

**Table 1 jcm-10-04869-t001:** Demographic and behavioral measurements; NC: cognitively normal participants; AD: mild cognitive impairment and early Alzheimer’s disease group; MMSE: Mini-Mental State Examination; TIV: total intracranial volume; SD: standard deviation.

	NC (*N* = 63)	AD (*N* = 63)	*z*	*p*-Value
Gender (m/f)	38/25	27/36	1.78	0.05
	Mean ± SD	Mean ± SD	*t*	*p*-value
Age (years)	74.54 ± 6.53	75.67 ± 7.92	−0.87	0.39
Education (years)	12.73 ± 2.55	12.90 ± 3.04	−0.34	0.73
MMSE score	29.25 ± 0.92	27.51 ± 2.04	6.20	<0.0001
TIV (cm^3^)	1474.23 ± 137.97	1480.96 ± 131.61	−0.28	0.78

**Table 2 jcm-10-04869-t002:** Grey matter voxels with significant correlation with (**a**) CVLT_1–5 and (**b**) CVLT_LDFR performance and their anatomical locations (Regions determined using MRIcron aal template) for the corrected and uncorrected data set, respectively. Different results in coordinates between the two data sets are displayed by numbers in brackets, whereas the coordinates for the uncorrected data set are enclosed. Missing values in cluster size and *p* value represent a voxel that appeared significant only in the corrected or uncorrected sample. Anatomical regions with asterisk (*) are undefined voxels in the aal template, and the regions are defined according to neighboring voxels.

	Anatomical Region	Side	MNI Coordinates Peak Voxel ^1^			Cluster Size (kE)		*p* (FWE-corr) ^2^	
			*x*	*y*	*z*	Corrected	Uncorrected	Corrected	Uncorrected
(**a**)	hippocampus *	right	32 (33)	−3	−24	55,074	57,166	<0.001	<0.001
	hippocampus *	left	−33	−7	−22	26,067	27,664	4.59 × 10^−12^	1.28 × 10^−12^
	precuneus	left	−5	−55	29	12,527	9555	3.91 × 10^−7^	7.36 × 10^−6^
	inferior temporal gyrus	right	50	−52	−12	3978	3850	0.006	0.007
	middle occipital gyrus	right	37	−71	34	9136	8816	1.20 × 10^−5^	1.62 × 10^−5^
	middle occipital gyrus *	left	−37	−87	38	2930		0.028	
	middle temporal gyrus	left	−53	−38	−5	9726	9385	6.42 × 10^−6^	8.81 × 10^−6^
(**b**)	hippocampus *	right	32	−3	−24	79,229	81,115	<0.001	<0.001
	amygdala	left	−27 (−28)	4	−20	55,166	56,963	<0.001	<0.001
	inferior parietal gyrus	right	26	−52	54	25,172	24,521	8.57 × 10^−12^	1.29 × 10^−11^
	precuneus	left	−4	−54	30	11,400	11,158	1.16 × 10^−6^	1.38 × 10^−6^
	inferior frontal gyrus, pars opercularis	left	−51	9	25	2694	2589	0.040	0.046
	medial frontal gyrus, pars orbitalis	left	−5	63	−1	10,499	10,347	2.82 × 10^−6^	3.14 × 10^−6^
	inferior parietal gyrus	left	−24	−66	44	6687		1.82 × 10^−4^	
	middle occipital gyrus	left	−30	−89	22		6434		2.38 × 10^−4^

^1^ MNI coordinates of best hit voxel in cluster. ^2^
*p* (FWE-corr) cluster-level.

## Data Availability

The data are not publicly available due to privacy restrictions. Data requests can be sent to the corresponding author.

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
