# Peer review of "Manual Correction of Voxel Misclassifications in Mesiotemporal Structures Does Not Alter Brain–Behavioral Results in an Episodic Memory Task"

_jcm, 2021, doi:10.3390/jcm10214869_

Round 1
Reviewer 1 Report
To date, there is obviously no gold standard as to the methodology of voxel-based morphometry (VBM) in cranial magnetic resonance imaging (MRI) assessment for diagnosing Alzheimer's disease (AD). Therefore, the manuscript is highly interesting. The authors' work is methodologically and statistically sound, with significant results being reported.
Some minor suggestions from the reviewer:
As far as the reviewer understands, the survey conducted among the authors' peers clearly explains the authors' motivation to conduct the study. To tell this interesting story even better, the reviewer recommends to place the results of the survey at the beginning of the manuscript, but not in the middle of the results section.
The reviewer suggests to include in the discussion section of the manuscript some thoughts as to cranial MRI preprocessing protocols for the assessment of structural brain alterations in other conditions. Does the authors' approach bear the potential to serve as a template for future gold standards in diagnosing structural brain alterations, e.g. in patients with chronic pain, spinal degeneration, or drug addiction? When assessing structural brain alterations in such conditions, are there any gold standards at all? Which limitations may occur?
A minor spell check is also recommended.
Other than that, the authors have to be commended for this work.
Reviewer 2 Report
Hartmann et al has provided evidence from126 human participants that manual correction of voxel misclassifications in mesiotemporal structures does not alter brain-behavioural results. The study design and data interpretation were appropriate. Ideally this should be validated by recruiting more human participants, but I guess it could be a follow up study.
I do not have any major concern for this manuscript, while minor grammatical errors were identified, e.g. line 55-57 is a bit hard to understand.
Reviewer 3 Report
The authors present a very interesting dataset comparing the laborous process of manually correcting grey matter voxels versus skipping this step in the preprocessing of automatic volumetry.
I commend the authors for this interesting work that demonstrates the lack of a uniform process in this regard and also suggests that it can be skipped entirely, which will certainly be appreacited by the community. I do have comments, however, regarding the selection and characterisation of patients included in this trial and the extent to which it can be generalized to the whole spectrum of alzheimer's disease, which is suggested in the conclusions.
Major comments:
1) How were patients diagnosed with dementia, and specifically, Alzheimer's dementia? At a mean MMSE of 26.5 it appears surprising that the DSM-IV criterion of significant impairment in functioning was met in all patients.
2) Along the same lines, how was MCI due to AD diagnosed, specifically, what kind of biomarker was used to establish AD pathology?
3) How long after diagnosis were MRIs acquired in the patient group?
4) The CVLT is rather specific for episodic memory / language function and thus it seems that the generalization to brain-behaviour is overstating the findings.
5) Generally, it appears that in patients with mild cognitive impairment or very early dementia, the manual correction of GM voxels did not lead to meaningfully different results than no correction with regards to its correlation with episodic memory. These are important results and will be appreciated by researchers. However, it seems that generalization to AD as a whole and brain-behavior as a field. It is unknown, and perhaps interesting to follow up on, how correction influences results in later stages of dementia when atrophy is more prominent. Please adapt the title and the conclusion accordingly.
Minor comments:
Figure 4 is hard to comprehend visually; perhaps provide substracted images, i.e. side by side comparison of the corrected and uncorrected segmentation.
